# A CONVERGENT FEDERATED CLUSTERING ALGORITHM WITHOUT INITIAL CONDITION

## ABSTRACT

Federated learning (FL) is a distributed learning paradigm that allows multiple clients to collaboratively train a shared model via communications to a central server. However, optimal models of different clients often differ due to heterogeneity of data across clients. In this paper, we address the dichotomy between heterogeneous models and simultaneous training in FL via a clustering structure among the clients. The clustering framework is one way to allow for high heterogeneity level between clients, while clients with similar data can still train a shared model. We define a new clustering framework for FL based on the (optimal) local models of the clients: two clients belong to the same cluster if their local models are close. We propose an algorithm, *Successive Refine Federated Clustering Algorithm* (`SR-FCA`), that treats each client as a singleton cluster as an initialization, and then successively refine the cluster estimation via exploiting similarity with other clients. In any intermediate step, `SR-FCA` uses an *error-tolerant* federated learning algorithm within each cluster to exploit simultaneous training and to correct clustering errors. Unlike some prominent prior works `SR-FCA` does not require any *good* initialization (or warm start), both in theory and practice. We show that with proper choice of learning rate, `SR-FCA` incurs arbitrarily small clustering error. Additionally, `SR-FCA` does not require the knowledge of the number of clusters apriori like some prior works. We validate the performance of `SR-FCA` on real-world FL datasets including FEMNIST and Shakespeare in non-convex problems and show the benefits of `SR-FCA` over several baselines.

## 1 INTRODUCTION

Federated Learning (FL), introduced in McMahan et al. (2016); Konečný et al. (2016); McMahan & Ramage (2017) is a large scale distributed learning paradigm aimed to exploit the machine intelligence in users' local devices. Owing to its highly decentralized nature, several statistical and computational challenges arise in FL, and in this paper, we aim to address one such challenge: heterogeneity.

The issue of heterogeneity is crucial for FL, since the data resides in users' own devices, and naturally no two devices have identical data distribution. There has been a rich body of literature in FL to address this problem of non-iid data. We direct the readers to two survey papers (and the references therein), Li et al. (2020a); Kairouz et al. (2019) for a comprehensive list of papers on heterogeneity in FL. A line of research assumes the *degree of dissimilarity* across users is small, and hence focuses on learning a single global model Zhao et al. (2018); Li et al. (2020b; 2019); Sattler et al. (2019); Mohri et al. (2019); Karimireddy et al. (2020). Along with this, a line of research in FL focuses on obtaining models personalized to individual users. For example Li et al. (2020b; 2021) use a regularization to obtain individual models for users and the regularization ensures that the local models stay close to the global model. Another set of work poses the heterogeneous FL as a meta learning problem Chen et al. (2018); Jiang et al. (2019); Fallah et al. (2020b;a). Here, the objective is to first obtain a single global model, and then each device run some local iterations (fine tune) the global model to obtain their local models. Furthermore Collins et al. (2021) exploits shared representation across users by running an alternating minimization algorithm and personalization. Note that all these personalization algorithms, including meta learning, work only when the local models of the users' are close to one another (see bounded heterogeneity terms $\gamma_H$ and $\gamma_G$ terms in Assumption 5 of Fallah et al. (2020b)).

On the other spectrum, when the local models of the users may not be close to one another, Sattler et al. (2021); Mansour et al. (2020); Ghosh et al. (2022) propose a framework of *Clustered Federated Learning*. Here users with dissimilar data are put into different clusters, and the objective is to obtain individual models for each cluster; i.e., a joint training is performed within each cluster. Among these, Sattler et al. (2021) uses a top-down approach using cosine similarity metric between gradient norm as

optimization objective. However, it uses a centralized clustering scheme, where the center has a significant amount of compute load, which is not desirable for FL. Also, the theoretical guarantees of Sattler et al. (2021) are limited. Further, in Duan et al. (2021), a data-driven similarity metric is used extending the cosine similarity and the framework of Sattler et al. (2021). Moreover, in Mansour et al. (2020), the authors propose algorithms for both clustering and personalization. However, they provide guarantees only on generalization, not iterate convergence. In Smith et al. (2017) the job of multi-task learning is framed as clustering where a regularizer in the optimization problem defines clustering objective.

Very recently, in Ghosh et al. (2022), an iterative method in the clustered federated learning framework called Iterative Federated Clustering Algorithm, or `IFCA`, was proposed and a *local convergence* guarantee was obtained. The problem setup for `IFCA` is somewhat restrictive—it requires the model (or data distribution) of all the users in the same cluster to be (exactly) identical. In order to converge, `IFCA` necessarily requires *suitable* initialization in clustering, which can be impractical. Furthermore, in Ghosh et al. (2022), all the users are partitioned into a fixed and known number of clusters, and it is discussed in the same paper that the knowledge about the number of clusters is quite non-trivial to obtain (see Section 6.3 in Ghosh et al. (2022)). There are follow up works, such as Ruan & Joe-Wong (2021), Xie et al. (2020), that extend `IFCA` in certain directions, but the crucial shortcomings, namely the requirements on *good* initialization and *identical* local models still remain unaddressed to the best of our knowledge.

In this paper, we address the above-mentioned shortcomings. We introduce a new clustering algorithm, Successive Refinement Federated Clustering Algorithm or `SR-FCA`, which leverages pairwise distance based clustering and refines the estimates over multiple rounds. We show that `SR-FCA` does not require any specific initialization. Moreover, we can allow the same users in a cluster to have non-identical models (or data distributions); in Section 2 we define a clustering structure (see definition 2.1) that allows the the same cluster models of the users to be different (we denote this discrepancy by parameter $\epsilon_1 (\geq 0)$. Furthermore, `SR-FCA` works with a different set of hyper-parameters which does not include the number of clusters and `SR-FCA` iteratively estimates this hyper-parameter.

**Clustering Framework and Distance Metric:** Classically, clustering is defined in terms of distribution from which the users sample data. However, in a federated framework, it is common to define a heterogeneous framework such as clustering in terms of other discrepancy metric; for example in Mansour et al. (2020), a metric that depends on the local loss is used.

In this paper, we use a distance metric across users' local model as a discrepancy measure and define a clustering setup based on this. Our distance metric may in general include non-trivial metric like Wasserstein distance, $\ell_q$ norm (with $q \geq 1$) that captures desired practical properties like permutation invariance and sparsity for (deep) neural-net training. For our theoretical results, we focus on strongly convex and smooth loss for which $\ell_2$ norm of iterates turns out to be the natural choice. However, for non-convex neural networks on which we run most of our experiments, we use a *cross-cluster loss* metric. For two clients $i, j$, we define their cross-cluster loss metric as the average of the loss of one client on the other's model, i.e., client $i$'s loss on the model of $j$ and the other way round. If this metric is low, we can use the model of client $i$ for client $j$ and vice-versa, implying that the clients are similar. We explain this in detail in Section 5. With the above discrepancy metric, we put the users in same cluster if their local models are close – otherwise they are in different clusters.

## 1.1 OUR CONTRIBUTIONS

**Algorithms and Technical Contribution.** We introduce a novel clustering framework based on local user models and propose an iterative clustering algorithm, `SR-FCA`. Note that, since the clustering is defined based on the optimal models of the users, we have no way to know the clustering at the beginning of the process. To mitigate this, we start with initially assigning a different cluster for each user, and run few local iterations of SGD/GD in parallel. We then form a clustering based on the pairwise distance between iterates. This clustering is refined (including merges/splits if necessary) over multiple rounds of our algorithm. We run federated training on each of the clusters to further improve the models. This step exploits collaboration across users in the same cluster. However, since there will be many mis-classifications while clustering based on iterates (which may be far from optimal models), ordinary federated training within clusters may lead to more errors (error propagation). To counter this we run a *robust* federated training (based on trimmed mean) within each clusters in the intermediate rounds, instead of straightforward federated learning. In particular, we use the first order gradient based robust FL algorithm of Yin et al. (2018) to handle the clustering error. Within a cluster, we treat the wrongly clustered users as outliers. However, instead of throwing the outliers away like Yin et al. (2018), we reassign them to their closest cluster.

When the loss is strongly convex and smooth, and $\mathsf{dist}(.,.)$ is $\ell_2$ norm, we show that, the mis-clustering error in the first stage of $\mathtt{SR\text{-}FCA}$ is given by $\mathcal{O}(md\exp(-n/\sqrt{d}))$ ( lemma 4.6), where $m$, $n$ and $d$ denote the number of users, the amount of data in each user and the dimensionality of the problem respectively. Moreover, successive stages of $\mathtt{SR\text{-}FCA}$ further reduce the mis-clustering error by a factor of $\mathcal{O}(1/m)$ (theorem 4.8), and hence yields arbitrarily small error. In practice we require very few refinement steps (we refine at most twice in experiments, see Section 5). Comparing our results with $\mathtt{IFCA}$ Ghosh et al. (2022), we notice that the requirement on the separation of clusters is quite mild for $\mathtt{SR\text{-}FCA}$. We only need the separation to be[1] $\tilde{\Omega}(\frac{1}{n})$. On the other hand, in certain regimes, $\mathtt{IFCA}$ requires a separation of $\tilde{\Omega}(\frac{1}{n^{1/5}})$, which is a much stronger requirement.

To summarize, a key assumption in any clustering problem (which is non-convex) is *suitable* initialization. However, $\mathtt{SR\text{-}FCA}$ removes this requirement completely by the above technique, and allows the clients to start arbitrarily. For our results, we crucially leverage (a) sharp generalization guarantees for strongly convex losses with sub-exponential gradients and (b) robustness property of the trimmed mean estimator (of Yin et al. (2018)).

As a by-product of $\mathtt{SR\text{-}FCA}$, we also obtain an appropriate loss minimizer for each cluster, defined in Eq. (2). We notice that the statistical error we obtain here is $\tilde{\mathcal{O}}(1/\sqrt{n})$. This statistical rate primarily comes from the usage of the robust estimator of Yin et al. (2018).

**Experiments.** We implement $\mathtt{SR\text{-}FCA}$ on wide variety of simulated heterogeneous datasets (rotated or inverted MNIST, CIFAR10) and real federated datasets (FEMNIST and Shakespeare Caldas et al. (2018)). With *cross-cluster* loss distance metric , we compare the test performance of $\mathtt{SR\text{-}FCA}$ with five baselines—(a) global (one model for all users) and (b) local (one model per user), (c)$\mathtt{IFCA}$, (d) CFL Sattler et al. (2021) and (e) Local-KMeans (local models clustered by KMeans on model weights). On simulated datasets, $\mathtt{SR\text{-}FCA}$ obtains test accuracy no worse than the best baseline and is able to recover the correct clustering. For CIFAR10, in particular, $\mathtt{SR\text{-}FCA}$ has $5\%$ better test accuracy than $\mathtt{IFCA}$. On real datasets, $\mathtt{SR\text{-}FCA}$ overcomes the issues faced by $\mathtt{IFCA}$, thereby beating it.

## 2 FEDERATED CLUSTERING AND OUR SETUP

In this section, we formally define the clustering problem. Let, $[n] \equiv \{1,2,...,n\}$. We have $m$ users (or machines) that are partitioned into disjoint clusters, denoted by the clustering map $\mathcal{C}^\star : [m] \to [C]$, where $C$ is the (unknown) number of clusters. Each user $i \in [m]$ contains $n_i \geq n$ data points $\{z_{i,j}\}_{j=1}^{n_i}$ sampled from a distribution $\mathcal{D}_i$. We define $f(\cdot;z) : \mathcal{W} \to \mathbb{R}$ as the loss function for the sample $z$, where $\mathcal{W} \subseteq \mathbb{R}^d$. Here, $\mathcal{W}$ is a closed and convex set with diameter $D$. We now define the population loss, $F_i : \mathcal{W} \to \mathbb{R}^d$, and its minimizer, $w_i^\star$, for each user $i \in [m]$: $F_i(w) = \mathbb{E}_{z \sim \mathcal{D}_i}[f(w,z)]$, $\quad w_i^\star = \min_{w \in \mathcal{W}} F_i(w)$. The original clustering $\mathcal{C}^\star$ is based on the population minimizers of users, $w_i^\star$. This is defined as:

**Definition 2.1** (Clustering Structure). For a distance metric $\mathsf{dist}(.,.)$, the local models satisfy

$$\max_{i,j:\mathcal{C}^\star(i)=\mathcal{C}^\star(j)} \mathsf{dist}(w_i^\star,w_j^\star) \leq \epsilon_1, \qquad \min_{i,j:\mathcal{C}^\star(i)\neq\mathcal{C}^\star(j)} \mathsf{dist}(w_i^\star,w_j^\star) \geq \epsilon_2. \tag{1}$$

where $\epsilon_1, \epsilon_2$, are non-negative constants with $\epsilon_2 > \epsilon_1$. This is illustrated in fig. 1.

The above allows the population minimizers inside clusters to be close, but not necessarily equal.

In practice, we have access to neither $F_i$ nor $w_i^\star$, but only the sample mean variant of the loss, the empirical risk, $f_i(w) = \frac{1}{n_i}\sum_{j=1}^{n_i} f(w,z_{i,j})$ for each user $i \in [m]$. Let $G_c \equiv \{i : i \in [m], \mathcal{C}^\star(i) = c\}$ denote the set of users in cluster $c$ according to the original clustering $\mathcal{C}^\star$. We can then define the population loss and its minimizer, per cluster $c \in [C]$ as $\mathcal{F}_c$

$$\mathcal{F}_c(w) = \frac{1}{|G_c|}\sum_{i \in G_c} F_i(w), \quad \omega_c^* = \operatorname*{argmin}_{w \in \mathcal{W}} \mathcal{F}_c(w) \tag{2}$$

Our final goal is to find a population loss minimizer for each cluster $c \in [C]$, i.e., $\omega_c^*$. To obtain this, we need to find the correct clustering $\mathcal{C}^\star$ and recover the minimizer of each cluster's population loss. There are two major difficulties in this setting: (a) the number of clusters is not known beforehand. This prevents us from using most clustering algorithms like $k$-means, and (b) The clustering depends on $w_i^\star$ which we do not have access to. We can estimate $w_i^\star$ by minimizing $f_i$, however when $n$, the minimum number of data points per user, is small, this estimate may be very far from $w_i^\star$.

---

[1]Here, $\tilde{\mathcal{O}}$ and $\tilde{\Omega}$ hide logarithmic dependence.

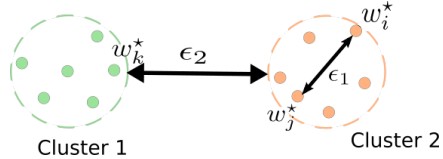 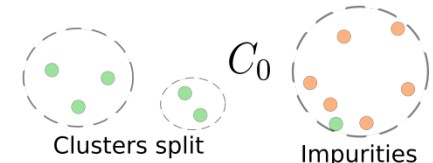

Figure 1: The dots represent the population risk minimizers for two clusters in dist(.,.) space according to $\mathcal{C}^\star$.

Figure 2: The dots represent the ERM in dist(.,.) space and the corresponding clustering $\mathcal{C}_0$ obtained after ONE_SHOT

The above difficulties can be overcome by utilizing federation. First, instead of estimating $w_i^\star$ for a single user, we can estimate $\omega_c^\star$, the population minimizer for each cluster, where nodes in the cluster collaborate to improve the estimate. Second, we can use these estimates of $\omega_c^\star$ to improve the clustering, according to definition 3.1.

## 3 ALGORITHM : SR-FCA

In this section, we formally present our clustering algorithm, SR-FCA. We first run the subroutine ONE_SHOT to obtain an appropriate initial clustering, which can be further improved. SR-FCA then successively calls the REFINE() subroutine to improve this clustering. In each step of REFINE(), we first estimate the cluster models for each cluster. Then, based on these models we regroup all the users using RECLUSTER() and, if required, we merge the resulting clusters, using MERGE(). We now explain the different subroutines in detail.

### 3.1 ONE_SHOT()

For our initial clustering, we create edges between nodes based on the distance between their locally trained models if $\text{dist}(w_i, w_j) \leq \lambda$, for a threshold $\lambda$. We obtain clusters from this graph by simply finding the connected components, which can be done in time linear in number of edges. We only keep the clusters which have at least $t$ nodes.

If our locally trained models, $w_{i,T}$, were close to their population minimizers, $w_i^\star$, for all nodes $i \in [m]$, then choosing a threshold $\lambda \in (\epsilon_1, \epsilon_2)$, we obtain edges between only clients which were in the same cluster in $\mathcal{C}^\star$. However, if $n$, the number of local datapoints is small, then our estimates of local models $w_{i,T}$ might be far from their corresponding $w_i^\star$ and we will not be able to recover $\mathcal{C}^\star$.

However, $\mathcal{C}_0$ is still a good clustering if it satisfies these requirements: (a) if every cluster in the range of the clustering map $\text{rg}(\mathcal{C}^\star) = [C]$ has a good proxy (in the sense of definition 3.1) in $\text{rg}(\mathcal{C}_0)$, and (b) each cluster in $\text{rg}(\mathcal{C}_0)$ has at most a small fraction ($< \frac{1}{2}$) of mis-clustered users in it. E.g., fig. 2 provides an example of one such good clustering when $\mathcal{C}^\star$ is defined according to fig. 1. We can see that even though $\mathcal{C}_0 \neq \mathcal{C}^\star$, the two green clusters and the single orange cluster in $\mathcal{C}_0$ are mostly "pure" and are proxies of Cluster 1 and Cluster 2 in fig. 1.

To formally define the notion of "purity" and "proxy", we introduce the notion of cluster label for any arbitrary clustering $\mathcal{C}'$, which relates it to the original clustering $\mathcal{C}^\star$.

**Definition 3.1** (Cluster label). We define $c \in [C]$, as the cluster label of cluster $c' \in \text{rg}(C')$ if the majority ($> 1/2$ fraction) of nodes in $c'$ are originally from $c$.

This definition allows us to map each cluster $c' \in \text{rg}(C')$ to a cluster $c$ in $\mathcal{C}^\star$ and thus define the notion of "proxy". In fig. 2, the cluster label of green clusters is Cluster 1 and that of orange cluster is Cluster 2. Further, using the cluster label $c$, we can define the impurities in cluster $c'$ as the nodes which did not come from $c'$. In fig. 2, the green node in orange cluster is an impurity. Based on these definitions, we can see that if clusters in $\mathcal{C}_0$ are mostly pure and can represent all clusters in $\mathcal{C}^\star$, then $\mathcal{C}_0$ is a good clustering.

### 3.2 REFINE()

We iteratively refine the clustering obtained by ONE_SHOT() using REFINE(). We describe the subroutines of a single REFINE step below. **Subroutine TrimmedMeanGD().** The main issue with ONE_SHOT(), namely, small $n$, can be mitigated if we use federation. Since $\mathcal{C}_0$ has atleast $t$ nodes per cluster, training a single model for each cluster will utilize $\geq tn$ datapoints, making the estimation more accurate. However, from fig. 2, we can see that the clusters contain impurities, i.e., users from a different cluster. To handle them, we use a robust training algorithm, TrimmedMeanYin et al. (2018).

### Algorithm 1: SR-FCA

**Input:** Threshold $\lambda$, Size parameter $t$
**Output:** Clustering $\mathcal{C}_R$
$\mathcal{C}_0 \leftarrow$ ONE_SHOT $(\lambda, t)$
**for** $r = 1$ to $R$ **do**
    $\mathcal{C}_r \leftarrow$ REFINE $(\mathcal{C}_{r-1}, \lambda)$
**end for**
ONE_SHOT $(\lambda, t)$
**for** all $i$ clients in parallel **do**
    $w_{i,T} \leftarrow$ Train local model for client $i$ for $T$ steps
**end for**
$G \leftarrow$ Graph with $m$ nodes and no edges
**for** all pairs of clients $i, j \in [m], i \neq j$ **do**
    Add edge $(i, j)$ to the graph $G$ if $\text{dist}(w_{i,T}, w_{j,T}) \leq \lambda$
**end for**
$\mathcal{C}_0 \leftarrow$ Connected components from graph $G$ with size $\geq t$
REFINE $(\mathcal{C}_{r-1}, \lambda)$
**for** all clusters $c \in \mathcal{C}_{r-1}$ **do**
    $\omega_{c,T} \leftarrow$ TrimmedMeanGD()
**end for**
$\mathcal{C}'_r \leftarrow$ RECLUSTER $(\mathcal{C}_{r-1})$
$\mathcal{C}_r \leftarrow$ MERGE $(\mathcal{C}'_r, \lambda, t)$

### Algorithm 2: RECLUSTER()

**Input:** Cluster models $\{\omega_{c,T}\}_{c \in \text{rg}(\mathcal{C}_r)}$, User models $\{w_i\}_{i=1}^m$, Clustering $\mathcal{C}_r$
**Output:** Improved Clustering $\mathcal{C}'_r$
**for** all nodes $i \in [m]$ **do**
    $\mathcal{C}'_r(i) \leftarrow \arg\min_{c \in \text{rg}(\mathcal{C}_r)} \text{dist}(w_i, \omega_{c,T})$
**end for**
**return** Clustering $\mathcal{C}'_r$.

### Algorithm 3: MERGE()

**Input:** Cluster models $\{\omega_{c,T}\}_{c \in \text{rg}(\mathcal{C}_r)}$, Clustering $\mathcal{C}'_r$, Threshold $\lambda$, Size parameter $t$
**Output:** Merged Clustering $\mathcal{C}_{r+1}$, Cluster models $\{\omega_{c,T}\}_{c \in \text{rg}(\mathcal{C}_{r+1})}$
$G \leftarrow$ Graph with nodes $\text{rg}(\mathcal{C}'_r)$ and no edges
**for** all pairs of clusters $c, c' \in \text{rg}(\mathcal{C}'_r), c \neq c'$ **do**
    Add edge $(c, c')$ to the graph $G$ if $\text{dist}(w_c, w_{c'}) \leq \lambda$
**end for**
$\mathcal{C}_{temp} \leftarrow$ Connected components from graph $G$ of size $\geq t$
For each cluster in $\mathcal{C}_{temp}$, merge the nodes of its component clusters to get $\mathcal{C}_{r+1}$
**for** $c \in \text{rg}(\mathcal{C}_{temp})$ **do**
    $G_c \leftarrow \{c' \in \text{rg}(\mathcal{C}'_r)$ which merged into $c\}$
    $\omega_{c,T} \leftarrow \frac{1}{|G_c|} \sum_{c' \in G_c} \omega_{c',T}$
**end for**
**return** $\mathcal{C}_{r+1}, \{\omega_{c,T}\}_{c \in \text{rg}(\mathcal{C}_{r+1})}$.

This subroutine is similar to FedAvg McMahan et al. (2016), but instead of taking the average of local models, we take the coordinate-wise trimmed mean, referred to as $\text{TrMean}_\beta$ where $\beta \in (0, 1/2)$ defines the trimming level.

**Definition 3.2** ($\text{TrMean}_\beta$). For $\beta \in [0, \frac{1}{2})$, and a set of vectors $x^j \in \mathbb{R}^d, j \in [J]$, their coordinate-wise trimmed mean $g = \text{TrMean}_\beta(\{x^1, x^2, \dots, x^J\})$ is a vector $g \in \mathbb{R}^d$, with each coordinate $g_k = \frac{1}{(1-2\beta)J} \sum_{x \in U_k} x$, for each $k \in [d]$, where $U_k$ is a subset of $\{x_k^1, x_k^2, \dots, x_k^J\}$ obtained by removing the smallest and largest $\beta$ fraction of its elements.

The full algorithm for TrimmendMeanGD is provided in Appendix B. Note that $\text{TrMean}_\beta$ has been used to handle Byzantine users, achieving optimal statistical rates Yin et al. (2018), when $< \beta$ fraction of the users are Byzantine. For our problem setting, there are no Byzantine users as such and we use $\text{TrMean}_\beta$ to handle users from different clusters as impurities.

Note the two requirements for good clustering $\mathcal{C}_0$ from ONE_SHOT: (a) if every cluster in $\mathcal{C}^\star$ has a proxy in $\mathcal{C}_0$, then the TrimmedMeanGD obtains at least one cluster model for every cluster in $\mathcal{C}^\star$, (b) if every cluster in $\mathcal{C}_0$ has a small fraction ($\beta < \frac{1}{2}$) of impurities, then we can apply $\text{TrMean}_\beta$ operation can recover the correct cluster model for every cluster.

We end up with a trained model for each cluster as an output of this subroutine. Since these models are better estimates of their population risk minimizers than before, we can use them to improve $\mathcal{C}_0$.

**Subroutine RECLUSTER().** The full algorithm for this subroutine is provided in algorithm 2. This subroutine reduces the impurity level of each cluster in $\mathcal{C}_0$ by assigning each client $i$ to its nearest cluster $c$ in terms of $\text{dist}(\omega_{c,T}, w_{i,T})$. Since $\omega_{c,T}$ are better estimates, we hope that the each impure user will go to a cluster with its actual cluster label. For instance, in fig. 2, the impure green node should go to one of the green clusters. If some clusters in $\text{rg}(\mathcal{C}^\star)$ does not have a good proxy in $\text{rg}(\mathcal{C}_0)$, then the nodes of this cluster will always remain as impurities.

**Subroutine MERGE().** We provide the full algorithm for this subroutine in algorithm 3. Even after removing all impurities from each cluster, we can still end up with clusters in $\mathcal{C}^\star$ being split, for instance the green clusters in fig. 2. In $\mathcal{C}^\star$, these form the same cluster, thus they should be merged. As these were originally from the same cluster in $\mathcal{C}^\star$, their learned models should also be very close. Similar to ONE_SHOT, we create a graph $G$ but instead with nodes being the clusters in $\mathcal{C}'_r$. Then, we add edges between clusters based on a threshold $\lambda$ and find all the clusters in the resultant graph $G$ by finding all connected components. Then, each of these clusters in $G$ correspond to a set of clusters in $\mathcal{C}'_r$, so we merge them into a single cluster to obtain the final clustering $\mathcal{C}_{r+1}$.

**Discussion.** SR-FCA uses a bottom-up approach to construct and refine clusters. The initialization in ONE_SHOT is obtained by distance-based thresholding on local models. These local models are improper estimates of their population minimizers due to small $n$, causing $\mathcal{C}_0 \neq \mathcal{C}^\star$. However, if $\mathcal{C}_0$ is not very bad, i.e., each cluster has $< \frac{1}{2}$ impurity fraction and all clusters in $\mathcal{C}^\star$ are represented, we can refine it.

REFINE() is an alternating procedure, where we first estimate cluster centers from impure clusters. Then, we RECLUSTER() to remove the impurities in each cluster and then MERGE() the clusters which should be merged according to $\mathcal{C}^\star$. Note that as these steps use cluster estimates which are more accurate, they should have smaller error. This iterative procedure should recover one cluster for each cluster in $\mathcal{C}^\star$, thus obtaining the number of clusters and every cluster should be pure, so that $\mathcal{C}^\star$ is exactly recovered. Computation and communication complexity of SR-FCA is deferred to Appendix A.

## 4 THEORETICAL GUARANTEES

In this section, we obtain the convergence guarantees of SR-FCA. For theoretical tractability, we impose additional conditions on SR-FCA. First, the $\mathsf{dist}(.,.)$ is the Euclidean ($\ell_2$). However, in experiments (see next section), we remove this restriction and work with other $\mathsf{dist}(.,.)$ functions. Here, we show an example where $\ell_2$ norm comes naturally as the $\mathsf{dist}(.,.)$ function.

**Proposition 4.1.** *Suppose that there are $m$ users, each with a local model $w_i^\star \in \mathbb{R}^d$ and its datapoint $(x,y_i) \in \mathbb{R}^d \times \mathbb{R}$ is generated according to $y_i = \langle w_i^\star, x \rangle + \epsilon_i$. If $x \sim \mathcal{N}(0, I_d)$ and $\epsilon_i \overset{i.i.d}{\sim} \mathcal{N}(0,\sigma^2)$, then $KL(p(x,y_i)||p(x,y_j)) = \mathbb{E}_x[KL(p(y_i|x)||p(y_j|x))] = \frac{d}{2\sigma^2}||w_i - w_j||^2$.*

Hence, we see that minimizing a natural measure (KL divergence) between the distributions for different users is equivalent to minimizing the $\ell_2$ distance of the underlying local models. This example only serves as a motivation, and our theoretical results hold for a strictly larger class of functions, as defined by our assumptions.

*Remark* 4.2 ($\lambda$ range). For the guarantees of this section to hold, we require $\lambda \in (\epsilon_1, \epsilon_2)$ and $t \leq c_{\min}$, where $c_{\min}$ is the minimum size of the cluster. We emphasize that, in practice (as shown in the experiments), we treat $\lambda$ and $t$ as hyper-parameters and obtain them by tuning. Hence, we do not require the knowledge of $\epsilon_1, \epsilon_2$ and $c_{\min}$.

The following assumptions specify the exact class of losses for which our analysis holds. Definitions provided in appendix G.

**Assumption 4.3** (Strong convexity). The loss per sample $f(w,.)$ is $\mu$-strongly convex with respect to $w$.

**Assumption 4.4** (Smoothness). The loss per sample $f(w,.)$ is also $L$-smooth with respect to $w$.

**Assumption 4.5** (Lipschitz). The loss per sample $f(w,.)$ is $L_k$-Lipschitz for every coordinate $k \in [d]$. Define $\hat{L} = \sqrt{\sum_{k=1}^d L_k^2}$.

We want to emphasize that the above assumptions are standard and have appeared in the previous literature. For example, the strong convexity and smoothness conditions are often required to obtain theoretical guarantees for clustering models (see Ghosh et al. (2022); Lu & Zhou (2016), including the classical $k$-means which assume a quadratic objective. The coordinate-wise Lipschitz assumption is also not new and (equivalent assumptions) featured in previous works Yin et al. (2018; 2019), with it being necessary to establish convergence of the trimmed mean procedure. Throughout this section, we require Assumption 4.3, 4.4 and 4.5 to hold.

**Misclustering Error** Since the goal of SR-FCA is to recover both the clustering and cluster models, we first quantify the probability of not recovering the original clustering, i.e., $C_r \neq C^\star$. Here and subsequently, two clusters being not equal means they are not equal after relabeling (see definition 3.1). We are now ready to show the guarantees of several subroutines of SR-FCA. First, we show the probability of misclustering after the ONE_SHOT step.

**Lemma 4.6** (Error after ONE_SHOT). *After running ONE_SHOT with $\eta \leq \frac{1}{L}$ for $T$ iterations, for the threshold $\lambda \in (\epsilon_1, \epsilon_2)$ and some constant $b_2 > 0$, the probability of error is $\Pr[\mathcal{C}_0 \neq \mathcal{C}^\star] \leq p \equiv md \ \exp(-n\frac{b_2\Delta}{\hat{L}\sqrt{d}})$, provided $\frac{n^{2/3}\Delta^{4/3}}{D^{2/3}\hat{L}^{2/3}} \gtrsim d$, where $\Delta = \frac{\mu}{2}(\frac{\min\{\epsilon_2 - \lambda, \lambda - \epsilon_1\}}{2} - (1 - \frac{\mu}{L})^{T/2}D)$ and $n = \min_{i \in [m]} n_i$.*

We would like to emphasize that the probability of error is exponential in $n$, yielding a *reasonable* good clustering after the ONE_SHOT step. Note that the best probability of error is obtained when $\lambda = \frac{\epsilon_1 + \epsilon_2}{2}$.

*Remark* 4.7 (Separation). In order to obtain $p < 1$, we require $\Delta = \Omega(\frac{\log m}{n})$. Since $\Delta \leq \frac{\mu}{2}\frac{\epsilon_2 - \epsilon_1}{4}$, we require $(\epsilon_2 - \epsilon_1) \geq \mathcal{O}(\frac{\log m}{n}) = \tilde{\mathcal{O}}(\frac{1}{n})$. Note that we require a condition only on the separation $\epsilon_2 - \epsilon_1$, instead of just $\epsilon_2$ or $\epsilon_1$ individually.

Although we obtain an exponentially decreasing probability of error, we would like to improve the dependence on $m$, the number of users. REFINE() step does this job.

**Theorem 4.8** (One step REFINE()). *Let* $\beta t = \Theta(c_{\min})$, *and* REFINE() *is run with* TrimmedMeanGD($\beta$). *Provided* $\min\{\frac{n^{2/3}\Delta'^{4/3}}{D^{2/3}}, \frac{n^2\Delta'^2}{\hat{L}^2\log(c_{\min})}\} \gtrsim d$, *with* $0 < \beta < \frac{1}{2}$, *where*

$\Delta' = \Delta - \frac{\mu B}{2} > 0$ *and* $B = \sqrt{2\hat{L}\epsilon_1/\mu}$. *Then, for any constant* $\gamma_1 \in (1, 2)$ *and* $\gamma_2 \in (1, 2 - \frac{\mu B}{2\Delta})$, *such that after running 1 step of* REFINE() *with* $\eta \leq \frac{1}{L}$, *we have*

$$\Pr[\mathcal{C}_1 \neq \mathcal{C}^\star] \leq \frac{m}{c_{\min}}\exp(-a_1 c_{\min}) + \frac{m}{t}\exp(-a_2 m) + (1 - \beta)m(\frac{p}{m})^{\gamma_1} + m(\frac{p}{m})^{\gamma_2} + 8d\frac{m}{t}\exp(-a_3 n\frac{\Delta'}{2\hat{L}})$$

*where* $c_{\min}$ *is the minimum size of the cluster. Further for some small constants* $\rho_1 > 0, \rho_2 \in (0, 1)$, *we can select* $\beta, \gamma_1$ *and* $\gamma_2$ *such that for large* $m, n$ *and* $\Delta'$, *with* $B << \frac{2\Delta'}{\mu}$, *we have* $\Pr[C_1 \neq C^\star] \leq \frac{\rho_1}{m^{1-\rho_2}}p$.

*Remark* 4.9 (Misclustering error improvement). Note that $\rho_2$ can be made arbitrarily close to 0 by a proper choice of $\gamma_1$ and $\gamma_2$. So, one step of REFINE() brings down the misclustering error by (almost) a factor of $1/m$, where $m$ is the number of users.

*Remark* 4.10 (Condition on $B$). Note that we require $B << \frac{2\Delta'}{\mu}$ for the above to hold. From the definition of $B$, when the intra-cluster separation $\epsilon_1$ is small, $B$ is small. So, for a setup like IFCA, where $\epsilon_1 = 0$, this condition is automatically satisfied.

Using single step improvement of REFINE, we obtain the improvement after $R$ steps of REFINE.

**Theorem 4.11** (Multi-step REFINE()). *If we run* $R$ *steps of* REFINE(), *resampling* $n_i$ *points from* $\mathcal{D}_i$ *and recompute* $w_i$ *as in* ONE_SHOT *for every step of* REFINE(), *then the probability of error for* SR-FCA *with* $R$ *steps of* REFINE() *is* $\Pr[\mathcal{C}_R \neq \mathcal{C}^\star] \leq \left(\frac{\rho_2}{m^{(1-\rho_1)}}p\right)^R$.

*Remark* 4.12 (Re-sampling). Although the theoretical convergence of Multi-step REFINE() requires resampling of data points in each iteration of REFINE(), we experimentally validate (see section 5, that this is not required at all.

*Remark* 4.13. Since each step of REFINE() reduces the probability of misclusteing by (almost) a factor of $1/m$, very few steps of REFINE() is often sufficient. In our experiments ( section 5), we need $1 - 2$ REFINE() steps.

**Convergence of cluster models:** SR-FCA also obtain an appropriate loss minimizer for each cluster.

**Theorem 4.14** (Cluster models). *Under the conditions described in theorem 4.8, after running* SR-FCA *for* $(R + 1)$ *steps of* REFINE(), *we have* $\mathcal{C}^{R+1} = \mathcal{C}^\star$ *and*

$$\|\omega_{c,T} - \omega_c^\star\| \leq (1 - \kappa^{-1})^{T/2}D + \Lambda + 2B, \text{where, } \Lambda = \mathcal{O}\left(\frac{\hat{L}d}{1 - 2\beta}\left(\frac{\beta}{\sqrt{n}} + \frac{1}{\sqrt{nc_{\min}}}\right)\sqrt{\log(nm\hat{L}D)}\right)$$

$\forall c \in \mathrm{rg}(C^\star)$, *with probability* $1 - \left(\frac{\rho_2}{m^{(1-\rho_1)}}p\right)^R - \frac{m}{c_{\min}}\frac{4du''}{(1 + nc_{\min}\hat{L}D)^d}$, *for some constant* $u'' > 0$.

## 5 EXPERIMENTS

We compare the performance of SR-FCA against several baselines on simulated and real datasets.

**Simulated Datasets:** We generate clustered FL datasets from MNIST LeCun & Cortes (2010) and CIFAR10 Krizhevsky et al. by splitting them into disjoint sets, one per client. For MNIST, by inverting pixel value, we create 2 clusters (referred to as inverted in table 1) and by rotating the image by $90, 180, 270$ degrees we get 4 clusters. Note that this is a common practice in continual learning Lopez-Paz & Ranzato (2017) and FL Ghosh et al. (2022). We set $m = 100, n = 600$. For CIFAR10, we create 2 clusters by rotating the images by 180 degrees and set $m = 32, n = 3125$. To emulate practical FL scenarios, we assume that only a fraction of the nodes participate in the learning procedure. For Rotated and Inverted MNIST, we assume that all the nodes participate, while for Rotated CIFAR10 50% of the nodes participate. For MNIST, we train a 2-layer feedforward NN, while for CIFAR10, we train a ResNet9 Page (2019). We train Rotated MNIST, Inverted MNIST and Rotated CIFAR10 for 250, 280 and 2400 iterations respectively with 2 refine steps for SR-FCA.

Table 1: Test Accuracy and standard deviations across 5 random seeds on simulated datasets. The highest accuracy is **bold**. `SR-FCA` is competitve with `IFCA` and beats it for Rotated CIFAR10.

| BASELINE | MNIST (INVERTED) | MNIST (ROTATED) | CIFAR (ROTATED) |
|---|---|---|---|
| SR-FCA | **92.03 ±0.30** | **91.66 ± 0.13** | **91.38 ± 0.27** |
| LOCAL | 76.52 ±0.54 | 85.55 ± 0.19 | 75.87± 0.33 |
| GLOBAL | 88.61 ± 0.77 | 80.88 ±1.55 | 88.75± 0.52 |
| CFL SATTLER ET AL. (2021) | 88.30 ± 1.12 | 80.47 ±0.44 | 87.59 ± 0.42 |
| LOCAL-KMEANS GHOSH ET AL. (2019) | 10.56 ± 1.31 | 10.35 ± 0.71 | 10.00 ±0.20 |
| IFCA GHOSH ET AL. (2022) | 91.55± 0.81 | **91.80 ± 0.25** | 86.05 ± 0.43 |

**Real Datasets:** We use two real federated datasets from leaf database Caldas et al. (2018). We sample $m = 50$ machines from FEMNIST and Shakespeare. FEMNIST is a Federated version of EMNIST with data on each client being handwritten symbols from a different person. Shakespeare is a NLP dataset where the task is next character prediction. For FEMNIST, train a CNN for while for Shakespeare we train a 2-layer stacked LSTM. For clustered FL baselines, we tune $K$, the number of clusters, with $K \in \{2,3,4,5\}$ for FEMNIST and $K \in \{1,2,3,4\}$. We run FEMNIST and Shakespeare for 1000 and 2400 iterations respectively and set number of refine steps to be 1 for `SR-FCA`.

We compare with standard FL baselines – Local (every client trains its own local model) and Global (a single model trained via FedAvg McMahan & Ramage (2017) on all clients). The main baseline we compare to is `IFCA`. Among clustered FL baselines, we consider CFL Sattler et al. (2021), which uses a top-down approach with cosine distance metric, and Local-KMeans Ghosh et al. (2019), which performs KMeans on the model weights of each client's local model. For real datasets, we compare with two additional baselines – FedSoft Ruan & Joe-Wong (2021) and `ONE_SHOT-IFCA` (initial clustering of `IFCA` obtained by `ONE_SHOT`), to assess if these variants can fix the issues of initialization in `IFCA`.

For `SR-FCA`, we tune the parameters $\lambda$ and $\beta$ for trimmed mean and set $t = 2$ and require at most 2 `REFINE` steps. We utilize the following metric based on cross-cluster loss which is better suited to measure distances between clients' models as these are neural networks.

**Definition 5.1** (Cross-Cluster distance). For any two clients $i, j \in [m]$, with corresponding local models $w_i$ and $w_j$ and local empirical losses $f_i$ and $f_j$, we define the cross cluster loss metric as $\mathsf{dist}_{\text{cross-cluster}}(w_i, w_j) = \frac{1}{2}(f_i(w_j) + f_j(w_i))$.

To extend this definition to distances between cluster $c$ and client $j$, such as those required by `REFINE`, we replace the client model $w_i$ and client loss $f_i$ by the cluster model $w_c$ and empirical cluster loss $f_c$ respectively. Similarly, to obtain distances between clusters $c$ and $c'$, which are required by `MERGE`, we replace the client models and losses by cluster models and losses respectively.

Further, for clustered FL baselines (`IFCA`, `CFL`, Local-KMeans, FedSoft, `ONE_SHOT-IFCA`) on real datasets, we tune the number of clusters. This is not required in `SR-FCA`.

**Test Metrics:** The test performance of any baseline is obtained by averaging over the clients, the test performance of each client on its model trained by the baseline. For the local baseline, it is the client's local model and for the global baseline it is the single global model. For `SR-FCA` and clustered FL baselines, it is the cluster model for the client. Note that we do not present convergence plots as different algorithms run in different number of stages.

For simulated datasets, the true clustering $\mathcal{C}^\star$ is known, so we report both the test accuracy and misclustering error in table 1 and 2 respectively. For real datasets, the true clustering is not known, therefore, we report only the final test accuracy in table 3. Further, we provide the test accuracy after `ONE_SHOT` and every `REFINE` step in table 4. The total time to run all experiments including hyperparameter tuning on a single NVIDIA-GeForce-RTX-3090 is 2 weeks.

## 5.1 RESULTS

Across all datasets, we find that `SR-FCA` is competitive with or outperforms all other algorithms in terms of both misclustering error and test accuracy.

**Comparison with Local and Global:** The Local algorithm has access to little data, while the Global model cannot handle the heterogeneity. Hence, as seen in tables 1 and 3, `SR-FCA` and other clustered FL baselines perform better as they find correct clusters with low heterogeneity inside each cluster.

Table 2: Average Misclustering error of clustered FL algorithms on test set across 5 random seeds for simulated datasets. The lowest error is **bold**. SR-FCA is competitve with IFCA and beats it for Rotated CIFAR10.

| BASELINE | MNIST (INVERTED) | MNIST (ROTATED) | CIFAR (ROTATED) |
|---|---|---|---|
| SR-FCA | **0.0** | **0.0** | **0.0** |
| CFL | 0.08 | 0.14 | 0.18 |
| LOCAL-KMEANS | 0.36 | 0.28 | 0.38 |
| IFCA | **0.0** | **0.0** | 0.50 |

Table 3: Test Accuracy and standard deviations across 5 seeds on Real datasets. Highest accuracy is **bold**. SR-FCA consistently outperforms IFCA.

| BASELINE | FEMNIST | SHAKESPEARE |
|---|---|---|
| SR-FCA | **83.83** $\pm$ 1.49 | **48.54 $\pm$ 0.69** |
| LOCAL | 66.18 $\pm$ 2.14 | 33.86 $\pm$ 1.22 |
| GLOBAL | 80.00 $\pm$ 3.02 | 45.28 $\pm$ 0.78 |
| CFL | 79.48 $\pm$ 3.48 | 44.14 $\pm$ 1.03 |
| IFCA | 81.93 $\pm$ 1.56 | 46.12 $\pm$ 1.22 |
| FEDSOFT | 78.74 $\pm$ 2.61 | 46.98 $\pm$ 1.25 |
| ONE_SHOT-IFCA | 81.62 $\pm$ 2.29 | 45.56 $\pm$ 1.15 |

**Comparison with CFL and Local-KMeans:** CFL and Local-KMeans use the cosine distance between gradients and $l_2$ distance between model weights which are not suitable for NN models. Local-KMeans performs the worst with $\approx 10\%$ test accuracy for simulated datasets. For real datasets, it's accuracy is $\leq 5\%$ so we do not report it. SR-FCA and IFCA use cross-cluster loss and client loss respectively, which are better suited to NN models, thus outperforming these baselines (see tables 1 and 3)

**Comparison with IFCA:** On **simulated datasets** (tables 1 and 2), we find that IFCA recovers $\mathcal{C}^\star$ and outperforms SR-FCA marginally for MNIST datasets. This is due to MNIST being a simpler and easier to learn dataset, even after adding heterogeneity via rotations or inversions. In contrast, for CIFAR10 the learning task is much more difficult, and IFCA, without proper initialization, ends up with all clients in only a single cluster after a few rounds resulting in a misclustering of 0.5, as seen in table 2. Thus it performs slightly worse than the global baseline in terms of test accuracy, as seen in table 1. From table 2, we see that SR-FCA correctly identifies $\mathcal{C}^\star$ and comprehensively beats IFCA in terms of test accuracy.

On **real datasets**, for IFCA, we need to find the correct number of clusters by tuning. For a random sample of clients, the true number of clusters might not be the same. SR-FCA can compute both the correct clustering and cluster models for every random sample, allowing it to beat IFCA, which fits the same number of clusters to every random sample. The difference is more pronounced for the more difficult Shakespeare dataset than the easier FEMNIST dataset. Further, the variants of FedSoft and ONE_SHOT-IFCA, have similar test performance to IFCA on real datasets. For FedSoft, which is a soft-clustering version of IFCA, the issue of initialization remains unresolved. Running IFCA after ONE_SHOT can only re-cluster the clients thus results in a similar performance. In short, SR-FCA outperforms IFCA as well as its variants.

**Intermediate Steps of SR-FCA:** For MNIST, multiple REFINE steps are necessary for best performance, however, for CIFAR10 and real datasets, only a single REFINE step achieves best performance (see table 4).

# 6 CONCLUSION

We conclude with a potential scope of improvement in SR-FCA. SR-FCA trains models obtained via TrimmedMeanGD() which assumes $\beta$ fraction of users inside each cluster are corrupted. Instead, if we run any federated optimization algorithm which can accommodate low heterogeneity, for instance FedProx Li et al. (2020b), inside each cluster in the final clustering $\mathcal{C}_R$, then we may obtain an improved error in theory ($\Lambda$ in theorem 4.14). We leave this as a future work.

| BASELINE | MNIST (INVERTED) | MNIST (ROTATED) | CIFAR10 (ROTATED) | FEMNIST | SHAKESPEARE |
|---|---|---|---|---|---|
| AFTER ONE_SHOT | 76.52 $\pm$ 0.54 | 85.55 $\pm$ 0.19 | 75.87 $\pm$ 0.33 | 66.18 $\pm$ 2.14 | 33.86 $\pm$ 1.22 |
| AFTER $1^{st}$ REFINE | 91.88 $\pm$ 0.39 | 91.63 $\pm$ 0.12 | **91.38 $\pm$ 0.27** | **83.83 $\pm$ 1.49** | **48.54 $\pm$ 0.69** |
| AFTER $2^{nd}$ REFINE | **92.03 $\pm$ 0.30** | **91.66 $\pm$ 0.13** | **91.38 $\pm$ 0.27** | **83.83 $\pm$ 1.49** | **48.54 $\pm$ 0.69** |

Table 4: Test Accuracies of intermediate steps of SR-FCA for all datasets

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
