# OpenReview forum: "A Convergent Federated Clustering  Algorithm without Initial Condition"
_ICLR.cc/2024/Conference — ICLR 2024 Conference Withdrawn Submission_

### Official Review · Reviewer_Vidh · 2023-10-31

**Soundness:** 2 fair
**Presentation:** 2 fair
**Contribution:** 2 fair
**Rating:** 3
**Confidence:** 4

**Summary:**

This paper introduces a federated clustering framework to address the dichotomy between heterogeneous models and simultaneous training in FL. This work proposes an algorithm, named SR-FCA, that treats each client as a singleton cluster as an initialization, and then successively refine the cluster estimation via exploiting similarity with other clients. The experimental results show that the proposed SR-FCA achieves a smaller clustering error and outperforms existing methods.

**Strengths:**

1. The paper introduces a clustering framework tailored for the Federated Learning environment, which aims to address the dichotomy between heterogeneous models and simultaneous training in FL.

2. The paper provides theoretical guarantees for the proposed method.

3. The experimental results appear to demonstrate that the proposed method (SR-FAC) achieves good performance in image classification tasks.

**Weaknesses:**

1. This paper could be improved in terms of clarity, such as simplifying the usage of symbols.

2. This paper aims to address the issue of collaborative training between clients in the case of model heterogeneity. However, there are some well-known solutions that have not been discussed yet, such as

[1] Tan, Yue, et al. "Fedproto: Federated prototype learning across heterogeneous clients." Proceedings of the AAAI Conference on Artificial Intelligence. Vol. 36. No. 8. 2022.
[2] Fang, Xiuwen, and Mang Ye. "Robust federated learning with noisy and heterogeneous clients." Proceedings of the IEEE/CVF Conference on Computer Vision and Pattern Recognition. 2022.
[3] Alam, Samiul, et al. "Fedrolex: Model-heterogeneous federated learning with rolling sub-model extraction." Advances in Neural Information Processing Systems 35 (2022): 29677-29690.

3. The method might not be suitable for all FL scenarios, especially when there's a vast number of clients or highly dispersed data.

4. While this study provides some theoretical analysis, it contains numerous assumptions which limit its applicability in the real world.

5. This study utilized only some small datasets for experiments. Incorporating commonly-used datasets like Tiny-Imagenet and CIFAR-100 in the experimental evaluation would be beneficial.

6. This study compares only a few methods in the performance comparison section and misses the most advanced ones from 2023. Consequently, it doesn't adequately illustrate the superiority of the proposed SR-FAC.

7. While this study discusses some works in the Introduction Section, it lacks a dedicated Related Work Section in its structure, failing to reflect a comprehensive review of the relevant literature.

**Questions:**

1. Why did the authors emphasize in Section 2 that the data size $n_i$ for each client $i$ is greater than $n$? Does $n$ have any special significance?

2. How does SR-FAC perform in terms of efficiency and scalability in large-scale data or node environments?

3. How does this work measure the distance between different architectural models?

4. How to determine the number of clusters? Is there any adaptive methods, which are particularly important for large-scale datasets?

5.This paper claims that the proposed method SR-FAC can overcome the issue of model heterogeneity, but this is not reflected in the experimental section.

6. How were some parameters in the experimental section selected, such as the optimizer and its associated parameters, training epochs, etc.?"

---

### Official Review · Reviewer_41D8 · 2023-10-31

**Soundness:** 2 fair
**Presentation:** 3 good
**Contribution:** 2 fair
**Rating:** 5
**Confidence:** 3

**Summary:**

The paper proposes a new clustering algorithm in federated learning named as SR-FCA. SR-FCA resolves several drawbacks existing in baseline IFCA. Most importantly, SR-FCA does not require any specific initialization, does not restrict all users in the same cluster to be exactly identical, and does not require knowledge of the cluster number apriori. Furthermore, this paper provides theoretical guarantees for the proposed SR-FCA.

**Strengths:**

- The paper has a strong motivation, e.g., provides a well-justified improvement over baseline in terms of arbitrary initialization, as well as no prior knowledge of cluster number and no restriction of the identical inner-cluster client models, which are all legitimate practical concerns.

- Experimental results seem to demonstrate that SR-FCA achieves convincing results compared to many baselines.

**Weaknesses:**

- The algorithmic novelty: the ingredients e.g., ONE_SHOT, MERGE, are techniques in many existing clustering approaches. Admittedly, they may be applied to centralized scenarios instead of federated settings as in this paper.

Nevertheless, the difference and unique challenges from centralized to federated settings may be more explicitly illustrated.

- The theoretical analysis: I generally find the theoretical analysis part is not difficult to follow compared to some exsiting literature like IFCA paper, and thus it is difficult to evaluate the actual theoretical contribution:

The assumptions used in this paper are relatively stronger, and thus less well justified, than IFCA. e.g., (1) the per-sample convexity and smoothness assumption, instead of the assumptions of population loss function as in IFCA, wondering whether assumption on individual loss function is necessary; (2) no characterization of stochasticity of the gradient, while existing literature like IFCA works with stochastic gradient, which is more realistic in deployment. wondering whether the stochastic gradient assumption poses extra complexity; (3) the coordinate-wise Lipschitz assumption is very restrictive, though the authors mention it is unavoidable when encountering trimmed mean procedure. Nonetheless, it is a very strong and unrealistic assumption in real world setting.

These assumptions need clearer justification for readers to evaluate the technical contribution of this part.

**Questions:**

Please see the questions raised in weaknesses

---

### Official Review · Reviewer_rp3L · 2023-11-02

**Soundness:** 2 fair
**Presentation:** 2 fair
**Contribution:** 2 fair
**Rating:** 5
**Confidence:** 2

**Summary:**

This paper proposes a new clustering framework for federated learning that allows for high heterogeneity level between clients while still enabling clients with similar data to train a shared model. The proposed algorithm, SR-FCA, does not require any good initialization and uses an error-tolerant federated learning algorithm within each cluster to exploit simultaneous training and correct clustering errors.

**Strengths:**

The paper proposes a novel federated clustering algorithm that addresses the challenges of heterogeneous models and simultaneous training in federated learning. The proposed algorithm introduces a clustering structure among the clients, which allows for high heterogeneity levels between clients while still enabling clients with similar data to train a shared model. The proposed algorithm incurs arbitrarily small clustering errors with proper choice of learning rate. The authors also demonstrate the convergence of the algorithm through experiments on simulated and real-world datasets.

**Weaknesses:**

1.	What if C0 is bad in initialization? How often will it happen?
2.	The author should provide more experiments and illustrations on several hyperparameters, especially for those thresholds (e.g., lambda, beta)
3.	What is the difference between a client and a node? Are they the same? It confused me during the reading.
4.	Is trimmedmean algorithm training anything? Does it produce an averaged model based on the local model within the cluster?

**Questions:**

Please refer to Strengths and Weaknesses.

---

### Official Review · Reviewer_TZiD · 2023-11-08

**Soundness:** 2 fair
**Presentation:** 1 poor
**Contribution:** 2 fair
**Rating:** 3
**Confidence:** 5

**Summary:**

The paper proposes a clustered FL method to tackle non-IID issues. Specifically, the proposed clustering method does not rely on selecting a good initialization.

**Strengths:**

1. The targeting problem is a practical issue of clustered FL.

**Weaknesses:**

1. The proposed algorithm is overly complicated. There are many simple ways to solve the initialization issue of clustering.

2. The selecting threshold of lambda is infeasible in calculating the distance between two high-dimensional vectors.

3. The paper's writing is confusing.

**Questions:**

Please refer to weakness.